# THE BELIEF STATE TRANSFORMER

**Edward S. Hu**[1,2]    **Kwangjun Ahn**[1]    **Qinghua Liu**[1]    **Haoran Xu**[1,3]    **Manan Tomar**[1,4]
**Ada Langford**[1]    **Dinesh Jayaraman**[2]    **Alex Lamb**[1]    **John Langford**[1]
[1]Microsoft Research    [2]University of Pennsylvania    [3]UT Austin    [4] University of Alberta

## ABSTRACT

We introduce the "Belief State Transformer", a next-token predictor that takes both a prefix and suffix as inputs, with a novel objective of predicting both the next token for the prefix and the previous token for the suffix. The Belief State Transformer effectively learns to solve challenging problems that conventional forward-only transformers struggle with, in a domain-independent fashion. Key to this success is learning a compact belief state that captures all relevant information necessary for accurate predictions. Empirical ablations show that each component of the model is essential in difficult scenarios where standard Transformers fall short. For the task of story writing with known prefixes and suffixes, our approach outperforms the Fill-in-the-Middle method for reaching known goals and demonstrates improved performance even when the goals are unknown. Altogether, the Belief State Transformer enables more efficient goal-conditioned decoding, better test-time inference, and high-quality text representations on small scale problems. Website: https://edwhu.github.io/bst-website

## 1  INTRODUCTION

Transformer models (Vaswani et al., 2017) have created a revolution in language modeling (Achiam et al., 2023) with the capability to generate language with many emergent properties at large scale. Examining these models for flaws in the pursuit of further progress, it's notable that they struggle with planning-heavy problems (Bubeck et al., 2023; Momennejad et al., 2024). How can we modify the architecture, objectives, and algorithms to create a model more capable of reaching goals?

To make progress, we propose the new Belief State Transformer (BST) architecture and objective in Section 2. Informally, a belief state is a sufficient amount of information from the past to predict the outcomes of all experiments in the future, which can be expressed as either a distribution over underlying world states or a distribution over future outcomes. The Belief State Transformer is similar to a standard decoder-only Transformer (*e.g.*, GPT2), except that it has encodings that run both forward and backward. Both of these encodings are fed into output heads which predict not only the next token after the prefix *but also* the previous token before the suffix as shown in Figure 1.

In Section 3 we then study in depth how the Belief State Transformer performs on a known-hard problem, the star graph (Bachmann & Nagarajan, 2024) which is an elegantly simple sequential prediction problem known to confound next token prediction approaches. It's easy to show that transformers can represent star graph solutions using known results (*e.g.*, (Sanford et al., 2024; Frydenlund, 2024)), so the problem here is one of optimization. In particular, we discover that parity problems can be embedded within star graph problems, with parity known as difficult for gradient-based optimizers. Surprisingly, despite throwing away the backward encoder for inference, the BST solves even relatively difficult instances of star graphs with experiments detailed in Section 3.4. Analyzing this discovery, the BST benefits from extra gradients, enabling avoidance of the parity-by-gradient problem systematically. We also show that data augmentation approaches and ablations of the BST cannot solve the star graph problem systematically.

Building on this discovery, Section 4 proves this is a general phenomenon: ideal Belief State Transformers recover the full belief state in a compact representation for the output head. In contrast, a forward-only transformer and even modifications which predict every future token do not. This result implies that the Belief State Transformer learns maximal information from a sequence—there is no other objective/representation which pulls more relevant information into a compact belief state.

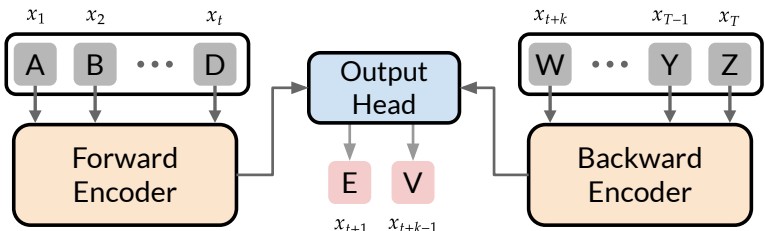

Figure 1: The Belief State Transformer has *two* encoders, one running forward and one backward with an output head for the next forward token and another for the previous backward token.

The Belief State Transformer creates new allowances which we further explore in Section 5. In particular, it's easy to specify a goal token (or tokens) explicitly to generate a goal-conditioned sequence. We compare this to the fill-in-the-middle approach (Bavarian et al., 2022), finding that the Belief State Transformer succeeds more effectively on the Tinystories dataset. The Belief State approach also enables inference planning because rollouts can occur with the Next head and a semi-independent evaluation of rollout quality can be enabled with the Prev head. Going further, we can use the Belief State Transformer as an embedder, with the relevant embeddings significantly superior to other transformer-based approaches.

Altogether, we show that the Belief State Transformer extracts more information (in theory and in practice) from a set of sequences, enabling Transformer models to perform well in new regimes. Performance at larger scale is of course an important question for further consideration.

## 2 THE BELIEF STATE TRANSFORMER

This section introduces the Belief State Transformer. We start by introducing the architecture and the training objective then discuss how to utilize the model for inference.

### 2.1 ARCHITECTURE AND OBJECTIVE

Let $x_{1:T}$ be shorthand for the sequence $x_1, ..., x_T$. First, we set up the following networks:

$$
\begin{array}{llll}
\text{Forward encoder} & \mathsf{F}(x_{1:t}) & \triangleright \text{ Encodes prefix} & \\
\text{Backward encoder} & \mathsf{B}(x_{t+k:T}) & \triangleright \text{ Encodes suffix} & \\
\text{Next decoder} & \hat{x}_{t+1} \sim \mathsf{T}_n(\cdot \mid \mathsf{F}(x_{1:t}), \mathsf{B}(x_{t+k:T})) & \triangleright \text{ Predicts next token} & (1) \\
\text{Prev decoder} & \hat{x}'_{t+k-1} \sim \mathsf{T}_p(\cdot \mid \mathsf{F}(x_{1:t}), \mathsf{B}(x_{t+k:T})) & \triangleright \text{ Predicts previous token} &
\end{array}
$$

The forward encoder aggregates the prefix into a latent $\mathsf{F}(x_{1:t})$, and the backward encoder aggregates the suffix into a latent $\mathsf{B}(x_{t+k:T})$. We use GPT2-style encoders throughout our experiments, including baselines. The output heads $\mathsf{T}_n$ and $\mathsf{T}_p$ then predict their respective tokens. In our experiments, the parameters of $\mathsf{T}_n$ and $\mathsf{T}_p$ are tied with only the last layer differing. See Figure 1 for an illustration.

The Belief State Transformer objective is the straightforward sum of the objectives of forward and backward Transformers conditioned on the prefix and suffix.

$$
\mathbb{E}_{t, x_{1:t}, k \leq T-t} \left[ \log \frac{1}{\mathsf{T}_n(x_{t+1} \mid \mathsf{F}(x_{1:t}), \mathsf{B}(x_{t+k:T}))} + \log \frac{1}{\mathsf{T}_p(x_{t+k-1} \mid \mathsf{F}(x_{1:t}), \mathsf{B}(x_{t+k:T}))} \right] \quad (2)
$$

An obvious alternative (called "Fill in the Middle" (Bavarian et al., 2022)) when both a prefix and suffix are available is simply putting them together and then using a forward encoder. Information-theoretically, Fill in the Middle works, but we'll see that the Belief State Transformer has several advantages: it causes the system to coalesce a compact belief state which has many benefits explored here. This approach also extracts $O(T^2)$ gradients from sequences enabling a gradient based optimizer to solve new problems as discussed in the next section. See Appendix E for code and scaling rules.

Training on all prefix-suffix pairs is surprisingly efficient. First, we cache all forward $f_{0:T} = \{\forall i \in [0:T] : \mathsf{F}(x_{1:i})\}$ and backwards $b_{1:T+1} = \{\forall i \in [1, T+1] : \mathsf{B}(x_{i:T})\}$ latents. Then the loss

Equation (2) is computed over training examples $(f_i, b_j)$ and their labels $(x_{i+1}, x_{j-1})$ for valid $i, j$ with $j - i > 1$. We first compute the gradients of the output decoder, and then add them up to compute the gradients of the encoders. Since there are $O(n)$ gradients over the output head per position of the forward or backward encoders this optimization saves a large amount of memory and compute.

## 2.2 BELIEF STATE INFERENCE

During inference time, the forward model $\mathsf{T}_n(\mathsf{F}(x_{1:t}), \mathsf{B}(\emptyset))$ is given a prefix $x_{1:t}$ and an empty suffix $\emptyset$. Autoregressive sampling (ARS) is always used, where we sample the next token $\hat{x}$ from the next token decoder, add it to the prefix, and repeat. Note that since $\mathsf{B}(\emptyset)$ can be precomputed, this approach requires no more parameters at inference time than a standard forward-only Transformer. Later in Section 5, we study more complex inference approaches.

## 3 TESTING PLANNING ABILITIES WITH STAR GRAPHS

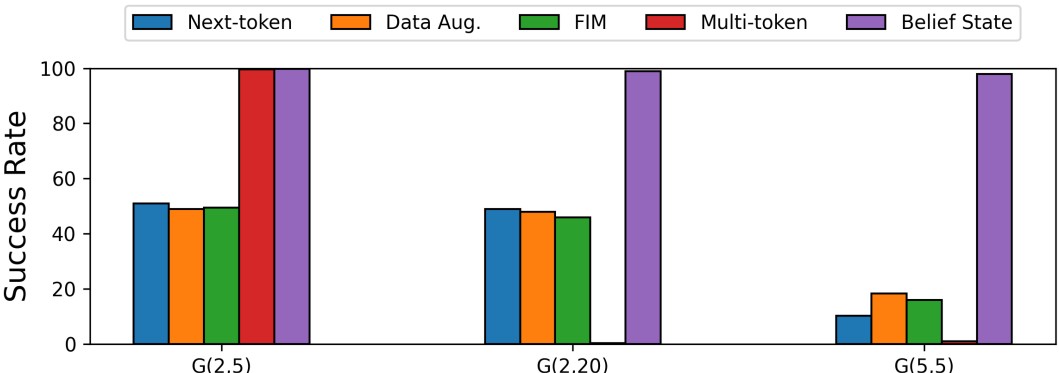

Figure 2: The Belief State Transformer outperforms baselines in all star graph navigation tasks.

Bachmann & Nagarajan (2024) propose the star graph problem as an elegantly simple yet challenging task for forward Transformers to solve. In Figure 2, we reproduce their results while adding a data augmentation baseline and the new Belief State Transformer results. Notably, the Belief State Transformer performs exceptionally well without relying on domain-specific adaptations. In the following sections, we explain the star graph problem, present a new theory to account for these results, provide a detailed discussion of our experiments, and ablate key design choices.

### 3.1 THE STAR GRAPH PROBLEM

A star graph (depicted in Figure 3) $G(d, l)$ is a graph with $d$ paths of length $l$ emanating out from the start node. To construct a graph, nodes $n_i$ are sampled uniformly from $\{1, \ldots, N\}$. A training example is formatted as a sequence containing the edge list $\mathcal{E}$, the start and end nodes, and a path of length $l$ from start to end: $[\mathcal{E} \mid n_1, n_l \mid n_1, n_2, n_3, \ldots n_l]$. Despite its simplicity, modern next token prediction models fail to solve it.

This task captures a core challenge found in practical planning tasks like story writing, where creating a coherent narrative requires the author to keep the story's resolution and backstory in mind while progressing through each plot point.

Figure 3: Illustration of the star graph problem from Bachmann & Nagarajan (2024).

### 3.2 WHY DO FORWARD-ONLY APPROACHES FAIL?

As shown by (Bachmann & Nagarajan, 2024, Appendix F.2) through extensive experiments, next-token predictors quickly learn a "*flawed cheat*" strategy: soon after training begins, forward-only transformers learn to arbitrarily select a neighbor of the current node since, aside from the start node,

each node has only one outgoing edge. This flawed strategy leads the model to choose a neighbor of the start node without accounting for the designated goal during inference, limiting the test accuracy to $1/d$, where $d$ is the number of neighbors of the start node. The key issue with this flawed cheat is that once the model learns it, finding the correct solution becomes exceedingly difficult. To better understand this challenge, we provide formal evidence of its difficulty for gradient optimization.

**Theorem 1** (Informal). *Once the "flawed cheat" strategy is perfected, learning the correct path is at least as difficult as learning full parity functions.*

A full proof is provided in Appendix B. The full parity problem is a notoriously hard problem for gradient-based optimizers (e.g., Shalev-Shwartz et al., 2017; Abbe & Sandon, 2023). In fact, it is conjectured that learning the full parity function requires exponentially many samples and computations in the input dimension (*e.g.*, Abbe & Boix-Adsera, 2022).

The analysis in this section indicates that once the flawed cheat is perfected, the limited supervision available for the task leads to significant intractability.

### 3.3 How does the Belief State Transformer Succeed?

The Belief State Transformer is trained to predict the previous token for every suffix, which prevents the problem from collapsing into the parity problem described in Theorem 1. At a high level, for our approach to reduce to the parity problem, a large number of gradients must approach zero. However, the Belief State Transformer ensures that the information necessary to construct path suffixes (e.g., $n_{2:l}$) is present in the input to the output head. From this information, predicting $n_2$ is straightforward, allowing the model to solve the problem effectively.

This is an instance of a more general phenomenon: the Belief State Transformer naturally converges toward extracting a compact, complete belief state as discussed further in Section 4.1.

### 3.4 Experimental Results on Star Graph

Here, we provide details on the Stargraph results in Figure 2. We run experiments on three types of graphs: $G(2,5)$, $G(5,5)$, $G(2,20)$. In each experiment, we choose one graph topology and generate many example graph sequences, with all methods receiving the same amount of data and every baseline receiving at least as much computation as the Belief State Transformer uses. For evaluation, the models are conditioned on the edge list, start, goal, and current path, with the task of next node prediction: $p(n_i \mid \mathcal{E}, n_1, n_l, n_{1:i-1})$. We report the path accuracy, which is the percentage of correct path generations during the test time, over 10,000 evaluation graphs.

**Empty suffix at inference.** The Belief State Transformer trains an additional encoder B for suffixes. However, as discussed in Section 2.2, we remove the dependency on B during inference time by pre-computing the backward latent $b_\emptyset = \mathsf{B}(\emptyset)$ with the end token and proceed with auto-regressive sampling of the Next decoder $\mathsf{T}_n(\cdot \mid \mathsf{F}(\mathcal{E}, n_1, n_T, n_1, \ldots n_i), b_\emptyset)$. The model is still able to produce goal-conditioned behavior since the goal node $n_T$ is present in the prefix input to the forward encoder.

**Baselines.** We select baselines that can be applied to a broad class of problems.

- **Forward-only next-token prediction:** This baseline follows the conventional strategy of training a Transformer with the next token prediction objective and teacher forcing, *i.e.*, the model is trained by feeding the correct previous token as input.
- **Data augmentation:** A common strategy to improve performance is to employ some form of data augmentation, although this requires some domain expertise to perform (Lee et al., 2024). This baseline augments the training data by replacing the goal with subgoals to potentially improve the learning of goal-conditioned behaviors. Specifically, the goal node, usually the terminal node in the path, is replaced with an intermediate node in the path. Then, a Transformer is trained with the next-token objective on this augmented dataset.
- **Fill-in-the-middle:** The FIM approach (Bavarian et al., 2022) moves a span of text from the middle to the end, and then trains the transformer with next token prediction. This can be seen as a generalization of the data augmentation approach, where subsequences of intermediate nodes are used in lieu of a single intermediate node.

- **Multi-token Prediction:** The multi-token approach refers to a Transformer trained to predict the entire path $\prod_{i=1}^{T} p(n_i \mid \mathcal{E}, n_1, n_T)$ in one forward pass without providing access to tokens from a partial path (hence the name "Teacherless" in Bachmann & Nagarajan (2024)).

Our code and baselines adopt the GPT2 Transformer architecture, using standard hyperparameter settings for number of layers, embedding dimension, etc. All models are trained on the same dataset with the same training budget. See Appendix C for additional training information and model setup.

## 3.5 RESULTS

As seen in Figure 2, the Belief State Transformer successfully learns to solve all the graphs. Since the Belief State Transformer uses an empty suffix, the parameter counts at inference time are very similar with minor variations driven by variations in output heads.

The baselines have varying degrees of success. The forward only baseline achieves at most a $1/d$ success rate, as it learns to output valid paths at random during inference time. Similarly, the data augmentation and FIM baselines perform poorly. We suspect that despite access to additional information like intermediate sub-goals or paths, the gradients which the model is exposed to are still inadequate to encourage the development of appropriate representations due to the easy availability of shortcut solutions.

The multi-token baseline, while successful in some smaller graphs like $G(2,5)$, fails to solve more complicated graphs with more arms ($G(5,5)$) or with longer path lengths ($G(2,20)$), reproducing prior results of Bachmann & Nagarajan (2024).

## 3.6 BELIEF STATE TRANSFORMER VARIANTS

Next, we ablate the Belief State Transformer to characterize its performance. The **Belief w/o Prev** ablation removes the previous token decoder and its objective from the training. The **Belief w/o Backward** ablation removes the backward encoder B from the training and inference process.

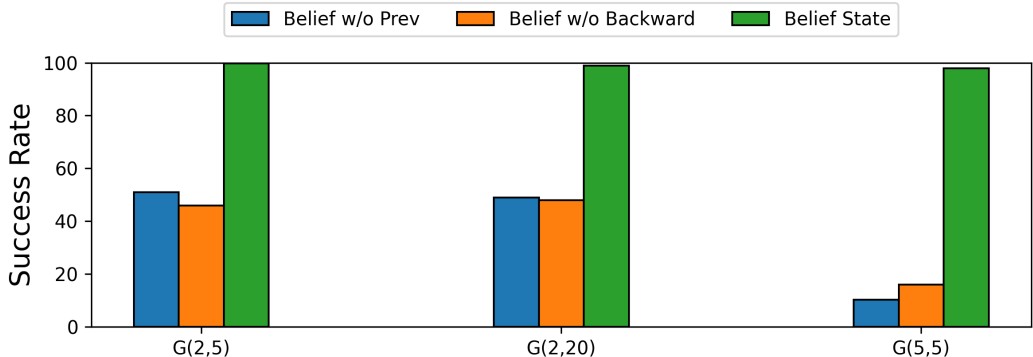

Figure 4: Ablations of the Belief State Transformer on the star graph. Both the belief state objective and the backward encoder are crucial.

The results in Figure 4 show that both the belief state objective and backward encoder are important components of the Belief State Transformer, and removing either drops performance back down to randomly guessing amongst valid paths. Prediction of the Prev token forces the Transformer to learn to represent long term dependencies which ends up being useful for goal-conditioned navigation.

## 4 BELIEF STATE TRANSFORMER ANALYSIS

In this section, we show that the Belief State Transformer discovers a compact belief state, and that the forward-only and multi-token approaches do not[1]. First, we formally define a belief state.

---

[1]The fact that the forward-only approach does not produce a compact belief state is well known. We formalize it here to contrast with the Belief State Transformer.

**Definition 1** (Belief State). *For any probability distribution over a set of sequences $P(x_{1:T})$, for any partial sequence $s = x_{1:t}$, a vector $v_s$ is a belief state for $s$ if there exists a randomized function $g(v_s)$ which can sample from the conditional distribution $\mathbb{P}(x_{t+1:T}|x_{1:t})$.*

By definition, a belief state captures all available information relevant for predicting the future tokens. Once the belief state is learned, there is no additional useful information to be gained—everything necessary for future predictions is already encoded within it.

## 4.1 BELIEF STATE DISCOVERY

We first show that successfully optimizing a Belief State Transformer results in a compact belief state.

**Theorem 2.** *Let $D = P(x_{1:T})$ represent any given probability distribution over a set of sequences. Consider an ideal Belief State Transformer that satisfies the following conditions for all prefixes $x_{1:t}$ and suffixes $x_{t+k+1:T}$:*

$$\mathsf{T}_n(x_{t+1} \mid \mathsf{F}(x_{1:t}), \mathsf{B}(x_{t+k+1:T})) = \mathbb{P}(x_{t+1} \mid x_{1:t}, x_{t+k+1:T}), \tag{3}$$

$$\mathsf{T}_p(x_{t+k} \mid \mathsf{F}(x_{1:t}), \mathsf{B}(x_{t+k+1:T})) = \mathbb{P}(x_{t+k} \mid x_{1:t}, x_{t+k+1:T}). \tag{4}$$

*Then, for any partial sequence $x_{1:t}$ supported by $D$, the forward encoding of the ideal Belief State Transformer, $\mathsf{F}(x_{1:t})$, is a belief state for $x_{1:t}$.*

*Proof.* Let $s_t = x_{1:t}$ denote the prefix. To prove that $f_t := \mathsf{F}(s_t)$ is a belief state for $s_t$, we must show that, given $f_t$, one can sample from the conditional distribution $\mathbb{P}(x_{t+1:T} \mid s_t)$.

The key observation is that the conditional distribution can be decomposed as follows:

$$\mathbb{P}(x_{t+1:T} \mid s_t) = \mathbb{P}(x_T \mid s_t)\mathbb{P}(x_{T-1} \mid s_t, x_T) \cdots \mathbb{P}(x_{t+1} \mid s_t, x_{t+2:T}).$$

For an ideal Belief State Transformer, this decomposition can be rewritten as:

$$= \mathsf{T}_p\big(x_T \mid f_t, \mathsf{B}(\emptyset)\big) \cdot \mathsf{T}_p\big(x_{T-1} \mid f_t, \mathsf{B}(x_T)\big) \cdots \mathsf{T}_p\big(x_{t+1} \mid f_t, \mathsf{B}(x_{t+2:T})\big),$$

Thus, using the forward encoding $f_t$, one can generate the remaining sequence $x_{t+1:T}$ by sampling in reverse order—first sampling $x_T$, then $x_{T-1}$ conditioned on $x_T$, and so on, until $x_{t+1}$. Each step involves using the Prev decoder and updating the backward encoder with the newly generated token.

Since the forward encoding $f_t$ enables sampling from the conditional distribution $\mathbb{P}(x_{t+1:T} \mid s_t)$ in this way, it follows that $f_t$ is a belief state for $s_t$. $\qquad\square$

## 4.2 NEXT-TOKEN OR MULTI-TOKEN PREDICTION DOES NOT GUARANTEE BELIEF STATES

Theorem 2 establishes that an ideal Belief State Transformer learns correct belief states. In contrast, the following two theorems demonstrate a fundamental limitation of standard next-token predictors and their variants when viewed from the belief state perspective.

**Theorem 3.** *Consider a standard next-token predictor with a forward encoder $\mathsf{F}$ and output head $\mathsf{T}$ such that*

$$\mathsf{T}(\mathsf{F}(x_{1:t})) = \mathbb{P}(x_{t+1} \mid x_{1:t}). \tag{5}$$

*There exists a distribution $P(x_{1:t})$ over sequences and a next-token predictor of the form Equation (5) such that the input to the output head is not a belief state.*

The proof is provided in Section B.2. Next, we analyze the case of multi-token training (see Section 3.4), where the model is asked to predict multiple future tokens at once.

**Theorem 4.** *Consider the multi-token setting where, for predicting $H$ tokens into the future, the model is of the form*

$$\mathsf{T}_j(\mathsf{F}(x_{1:t})) = \mathbb{P}(x_{t+j} \mid x_{1:t}) \quad for\ j = 1, 2, \ldots, H. \tag{6}$$

*There exists a distribution $D = P(x_{1:t})$ over sequences and a multi-token model satisfying Equation (6) such that the input to the output head is not a belief state.*

The proof is provided in Section B.3. In summary, the analysis in this section highlights the inherent limitations of next-token predictors: the input to the output head fails to capture the belief state.

## 5 EXPERIMENTING WITH THE BELIEF STATE TRANSFORMER

Given a model jointly trained on prefixes and suffixes, the representation is useful in new ways which are not available to simple forward Transformers. Here we detail two different forms of search based on forward and backward probabilities (respectively) as well as belief state embedding extractions.

**Setup.** We use TinyStories (Eldan & Li, 2023), a dataset consisting of synthetic short stories. TinyStories aims to represent key challenges in text generation while keeping training tractable for small to medium scale models. We tokenize the dataset into a vocabulary space of size 1000, and discard stories greater than 256 tokens long resulting in a dataset consisting of 2.7 million stories.

During evaluation, the models generate text using prefix-suffix snippets from an evaluation set of 100 unseen stories. Given stories from two competing models, GPT4 is then asked to output an analysis of each story examining multiple factors (e.g. grammar, flow, cohesiveness, creativity) before outputting a final recommendation. We follow best practices in evaluating with multiple trials and shuffling choice order. We report the winrate and confidence interval (CI) for each model. See Appendix D.2 for more details and examples of the GPT4 judge outputs and scoring.

### 5.1 GOAL-CONDITIONED TEXT GENERATION

In the goal-conditioned setting, the user provides the model with a prefix and suffix, and the model infills the text in between. See below for an example of the prefix and suffix. We describe a goal-conditioned planning procedure with the Belief State Transformer for text generation in Algorithm 1.

**Method.** The algorithm performs $n$ roll-outs. In each roll-out (starting at line 3), a candidate trajectory is generated that differs from the greedy trajectory but maintains high probability. The key features of the process are as follows:

1. *Greedy generation*: Starting with a sequence $s$ popped from the priority queue $Q$, a candidate trajectory is extended up to a maximum length $k$ using argmax greedy selection (line 6). The completed trajectory is appended to the set of candidates $C$ (line 11).

2. *Priority queue update for future generation*: Simultaneously, the priority queue $Q$ is updated to track alternative candidate sequences. This is done by appending non-greedy tokens to the queue, with their priority set by the *relative suboptimality* of each token. Specifically, lines 8 and 9 add alternative tokens with priority equal to the current sequence priority

---

**Algorithm 1** Goal-conditioned Planning

**Require:** prompt $x_{1:t}$, goal $x_{t+k:T}$, horizon $k$, rollouts $n$

1: Priority Queue $Q \leftarrow (1, x_{1:t})$
2: Candidates $C \leftarrow \emptyset$
   *Roll-outs start here*
3: **for** $j = 1, \ldots, n$ **do**
4:   (priority $r$, sequence $s$) $\leftarrow$ pop($Q$)
5:   **while** $|s| < t + k - 1$ **do**
6:     *Greedy generation*
      $x_{\max} \leftarrow \arg\max_x \mathsf{T}_n(x \mid \mathsf{F}(s), \mathsf{B}(x_{t+k:T}))$
7:     *Priority queue update for future generation*
      value $\mathsf{T}_{\max} = \mathsf{T}_n(x_{\max} \mid \mathsf{F}(s), \mathsf{B}(x_{t+k:T}))$
8:     **for** $x \neq x_{\max}$ **do**
9:       $Q \leftarrow (r \cdot \frac{\mathsf{T}_n(x \mid \mathsf{F}(s), \mathsf{B}(x_{t+k:T}))}{\mathsf{T}_{\max}}, s + x)$
10:     $s \leftarrow s + x_{\max}$
11:   *Appending $C$ with greedy generation*
    $C \leftarrow C \cup \{s + x_{t+k:T}\}$
12: *Scoring*
  **Output:** $\arg\max_{x_{1:T} \in C}$ of Equation (7)

---

multiplied by the ratio of the alternative token's probability to that of the greedy token. Since this ratio is $\leq 1$, priorities decreases with suboptimality. Also, this ensures that partial sequences of different lengths remain comparable, as suboptimality is independent of sequence length. This encourages branching at ambiguous points.

3. *Scoring*: Once candidate trajectories are generated, they are scored by evaluating the consistency of generated tokens $x_{t+1:t+k-1}$ with the goal $x_{t+k:T}$ using the next-head probability:

$$\prod_{i=t+k}^{T} \mathsf{T}_n(x_i \mid \mathsf{F}(x_{1:i-1}), \mathsf{B}(x_{i+1:T})) \tag{7}$$

We then return the highest-scoring trajectory based on Equation (7).

**Baseline.** We select the Fill-in-the-middle (FIM) (Bavarian et al., 2022) approach as a natural goal-conditioned baseline. FIM trains a single forward-only transformer where the input is the

concatenated suffix and prefix, and the objective is next token prediction. Because the forward-only approach is radically less efficient, we sample suffixes uniformly at random during training. During inference, beam search (Graves, 2012) is performed to search for high-probability sequences.

For fair comparisons, we train the FIM and Belief State Transformer with a similar amount of resources in terms of wall-clock and GPU usage, and set the architecture configuration so that the number of parameters between models is similar (80M for Belief, 85M for FIM). During inference time, we configure the search methods so that the computational budget is similar, i.e. setting the number of beams and depth to the values we use in Algorithm 1. See Appendix D.1 for more details.

**Results.** As seen in Figure 5, GPT4 prefers the Belief State transformer 64% of the time with a binomial confidence interval of $[54\%, 74\%]$. Qualitatively, the stories of Belief State Transformer are superior to FIM in several ways. The Belief State outputs more frequently display clear narrative structure, with a beginning, middle, and end, while the FIM model tends to generate shorter stories with simpler structure and repetitions. The FIM model also frequently failed to connect its completions to the suffix, resulting in abrupt and grammatically incorrect transitions. See below for an example.

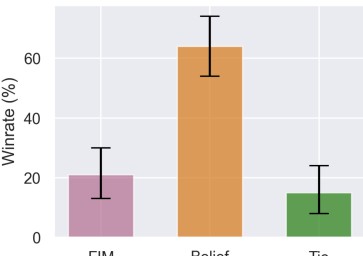

Figure 5: Storytelling winrates.

**Prefix:** *Once upon a time, there was a king. He was a big and strong king who ruled over his kingdom. One day, he wanted to take a nice and long bath, so he filled the bathtub with warm water*

**Belief:** *. He got in and started to wash. The king was very busy, but he was so busy that he didn't notice the water was getting too hot. Suddenly, he felt a sharp pain in his foot. He looked down and saw it burning. He was very scared and he quickly put his foot in the water. The king was very worried. He*

**FIM:** *and lots of bubbles. He stepped into the bathtub and started to play with the bubbles. Suddenly, the king noticed that one of the bubbles was getting bigger and bigger. He asked the king, "What's happening?" The king replied, "The bubble is getting bigger and bigger. It's getting*

**Suffix:** *quickly grabbed a cloth and began to clean it up. The king got so hot from cleaning up the mess that he decided to take another soak in the bathtub. He put a lot of bubbles in the water to make it nice and bubbly. He relaxed again and felt all the worries wash away.*

In this example, the Belief State Transformer's story has a clear narrative structure with setup, conflict and resolution. It is also able to correctly connect the prefix, generated text, and suffix. On the other hand, the FIM model has repetitions ("bigger and bigger"), confusing dialogue where a previously unmentioned person asks the king a question, and an abrupt and grammatically incorrect transition to the suffix ("It's getting quickly grabbed"). See Appendices D.2 and D.3 for more examples as well as the grade reports and evaluation outputs of the GPT4 judge.

## 5.2 UNCONDITIONAL TEXT GENERATION

**Method.** Next, we investigate the unconditional setting, or when the goal is unknown. We reuse the logic in Algorithm 1 with two following changes: setting the goal to the empty input $z'_\emptyset := \{\emptyset\}$, and scoring the sequences with the previous head $\mathsf{T}_p$ over a fixed amount of suffix tokens $k$. We choose to use the previous head rather than the next head as a semi-independent evaluator of next-generated tokens to reduce the bias associated with self-evaluation. By scoring over the last $k$ tokens, this selects for trajectories whose endings are more likely.

$$\prod_{i=T-k}^{T} \mathsf{T}_p(x_i \mid \mathsf{F}(x_{1:i-1}), \mathsf{B}(x_{i+1:T}))$$

(8)

**Results.** As seen in Figure 6, the Belief State transformer outperforms the FIM model. The Belief State Transformer outputs consistently have correct grammar, whereas the FIM models often abruptly end in the middle of sentences. This grammatical flaw is consistently picked up by the GPT4 judge, and results in the FIM's lower winrate. See below for an example.

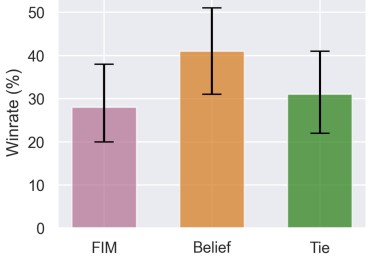

Figure 6: Unconditioned winrates.

> **Prefix:** *You don't have to be scared of the loud dog, I'll protect you". The mole felt so safe with the little girl. She was very kind and the mole soon came to trust her. He was*
>
> **Belief:** so gentle and kind. The little girl was so happy to have a new friend. She wanted to show the mole all her favorite things. She took him to the park and showed him all the different things she could see. The mole was so excited to see the different things. He ran around and played with the other kids. He even found a ball and a toy car. The little girl was so happy to see the mole having fun.
>
> **FIM:** always there to protect her. One day, the mole was walking through the woods when he heard a loud noise. He looked up and saw a big, scary dog. The dog was barking and growling at the mole. The mole was so scared that he ran away as fast as he could. The dog chased after the mole, but the mole was too fast. The dog was getting closer and clos
>
> **Suffix:** ∅

## 5.3 ABLATIONS

Finally, we conduct a few ablations to verify our design choices. First, the **Belief Beamsearch** ablation uses beamsearch to generate the candidate set rather than the priority queue scheme in Algorithm 1. Next, in the unconditioned setting, the **Belief Next Score** ablation uses the next head $\mathsf{T}_n$ to score the suffix instead of $\mathsf{T}_p$.

Figure 7 shows the performance drops in both cases. The **Belief Beamsearch** outputs is more prone to repetition than Algorithm 1. Next, the **Belief Next Score** ablation frequently outputs stories that end abruptly and incorrectly, similar to the FIM baseline in the unconditional setting.

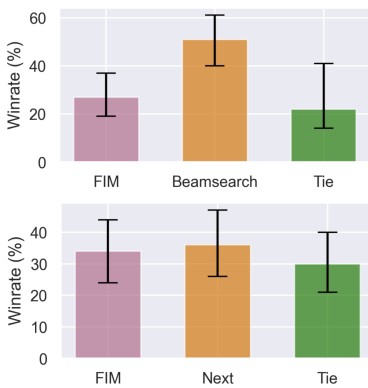

Figure 7: Ablations.

## 5.4 UNDERSTANDING BELIEF STATES

We investigated the representations learned by the Belief State Transformer to provide insight into what the states capture and how they are learned. First, we verified that the representations are indeed belief states, by training small MLP networks from the first hidden state on the G(2,5) Stargraph to predict the future token on a specific timestep. We found that the Belief State Transformer has the information to predict all future tokens, while the information contained within the state learned using the Forward-Only objective is incomplete (Figure 8, top-left).

An analysis of how the probe accuracy improved over the course of training revealed a clear trend. The belief state captures information about the tokens at the end of the sequence at the beginning of training, while information about the earlier tokens is learned later (Figure 8, top-right).

The same probing technique can understand how well the Belief State successfully captures information about the graph description. We found that the Belief State Transformer captures more information about the graph description than the Forward-Only model (Figure 8, bottom). Capturing the graph description completely is sufficient but not necessary for learning the belief state, since all the information in the graph description might not be used for predicting the future tokens.

## 6 OTHER RELATED WORK

Many previous works explore non-left-to-right approaches for sequence modelling. ELMo (Peters et al., 2018) and BERT (Devlin et al., 2018) were prominent approaches that trained language representations with past and future information. Gu et al. (2019) model the generation order as latent variables and performs beam search over the generation order itself to produce the final text. This is beneficial because in general, the best ordering for decoding is task-dependent. On the other hand, Welleck et al. (2019) explore generating text in a binary tree order, but their model struggles to outperform traditional left-to-right autoregressive generation in tasks like language modeling, machine translation, sentence completion, and word reordering. In Nguyen et al. (2024), they propose training two separate transformers and implement several strategies to "meet in the middle" during decoding. In contrast, our approach involves jointly training a transformer capable of both forward and backward decoding, which has important implications for creating compact belief states. A general study of the quality of backward prediction is done in Papadopoulos et al. (2024) discovering that backward prediction is possible but generally slightly worse than forwards

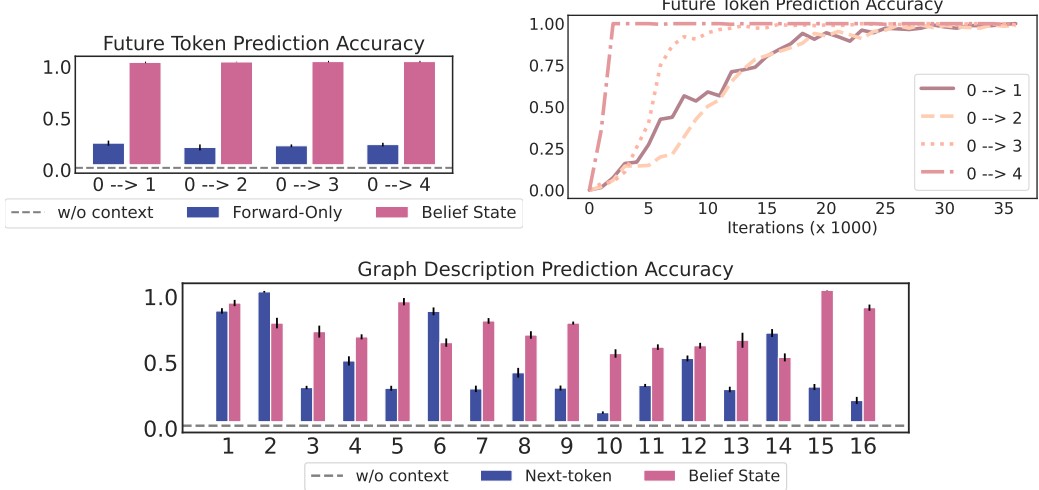

Figure 8: **Probing results on star graphs** show that Belief State Transformer's learned representation captures more information about future tokens (top-left). Top right: this belief state develops earlier in training for tokens which are further in the future. Bottom: besides future tokens, the inputted graph descriptions are also captured in the embeddings.

prediction. Somewhat further afield, combined forward/backward decoding approaches with RNNs have proved useful (Mou et al., 2015; Serdyuk et al., 2018; Mangal et al., 2019) and similarly for neural machine translation (Liu et al., 2016; Zhang et al., 2018).

The concept of predicting multiple tokens has been explored in other works. Gloeckle et al. (2024); Cai et al. (2024) aim to increase decoding speed by employing models that predict multiple tokens rather than just the next token. DeepSeek-V3 (Liu et al., 2024) pretrains the LLM to predict the next two tokens, and shows the objective is beneficial in the large scale setting. Such approaches align with our theoretical insights that predicting multiple tokens fosters the creation of more robust representations of future contexts.

Shai et al. (2024) show that transformers create a non-compact representation of belief states within their residual stream. Compact belief states are accessible in state-space models (Hasani et al., 2020; Gu & Dao, 2023), although this comes with different trade-offs compared to transformer-based approaches. For example, the Mamba training process is known to fail on star graphs (Bachmann & Nagarajan, 2024).

## 7 CONCLUSION

The Belief State Transformer advances goal-conditioned next-token predictors, effectively addressing the limitations of traditional forward-only transformers. The ability of our model to learn a compact belief state provides a maximal solution: since the belief state contains *all* the information useful in predicting the future, no more complex objective can elicit more information. Through experiments with both star graphs and story writing tasks, we have demonstrated the necessity of each component of our model, particularly in challenging scenarios where standard transformers struggle.

Looking ahead, while our results demonstrate the superior performance of the Belief State Transformer in small scale story writing, exploring its application to other goal-conditioned tasks would be valuable. Our current experiments serve as proof-of-concept. Further investigation into the scalability of our approach to larger practical scenarios is essential.

## 8 ACKNOWLEDGEMENTS

This work was supported in part by the NSF CAREER 2239301, NSF 2331783, and DARPA TIAMAT HR00112490421 grants. The authors would like to thank the MSR AI Frontiers team for fruitful discussions and support, and the ICLR reviewers for their constructive feedback.

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

## A    STATEMENT OF CONTRIBUTIONS

Edward Hu developed the BST, set up the code and infrastructure, proposed and ran the experiments. Kwangjun Ahn developed the BST, assisted with the Stargraph experiments, and contributed to theoretical analysis. Qinghua Liu contributed the parity proof and gave feedback to the project. Haoran Xu assisted in debugging, ran baselines, and gave feedback. Manan Tomar contributed early architecture code, ran the probe analysis, and gave feedback. Ada Langford contributed code for the Stargraph experiment. Dinesh Jayaraman advised EH and gave feedback. Alex Lamb oversaw the project, developed the BST, debugged models, proposed experiments, and ran the probe analysis. John Langford oversaw the project, developed the BST, contributed to BST theory, proposed experiments, and assisted with writing.

## B    PROOFS OF THEORETICAL RESULTS

### B.1    PROOF OF THEOREM 1

In this section, we provide a formal statement and a proof of Theorem 1.

As discussed in the main text, the starting point of our argument is the observation by (Bachmann & Nagarajan, 2024, Appendix F.2) that the model quickly learns the aforementioned "flawed cheat" solution, which prevents it from learning the true solution. Specifically, next-token predictors rapidly adopt this flawed shortcut, but make little progress in correctly predicting the first vertex on the path. Below, we provide formal evidence that learning the flawed cheat indeed hinders the model from learning the true solution.

In essence, once the flawed cheat solution is perfected, the model receives supervision only from the prediction of the first vertex after the starting node. This effectively reduces the star graph task to the following simplified task:

**Task 1.** *Given a stargraph and a goal node, predict the correct neighbor of the starting node.*

The formal statement of Theorem 1 is that Task 1 is at least as difficult as learning the full parity function (i.e., predicting whether the sum of the elements in a binary string is even or odd).

**Theorem 5.** *For every full parity problem, there exists a stargraph such that solving Task 1 provides a solution to the parity problem.*

Notably, empirical evidence shows that gradient-based methods struggle to learn the full parity function in standard training setups, especially in high dimensions (e.g., Shalev-Shwartz et al., 2017; Abbe & Sandon, 2023). It has been also conjectured that learning the full parity function requires a number of samples and computations exponential in the input dimension (e.g., Abbe & Boix-Adsera, 2022). Consequently, Task 1 inherits this difficulty, as it encapsulates learning the full parity function as a special case.

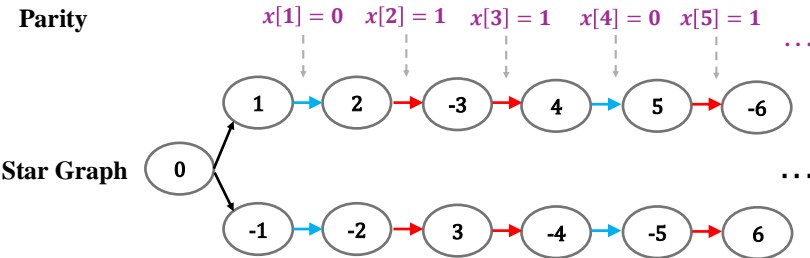

Figure 9: Reduction from learning parity to learning star graph. Note that if $\text{parity}(x) = 0$ (even), then the shortest path from center $0$ to vertex $n$ passes $1$. Otherwise, it passes $-1$. As a result, solving the star graph instance induced by $x$ implies correctly predicting the parity of $x$.

**Proof of Theorem 5.** Consider a star graph with $2n+1$ vertices. For simplicity, we index the vertices by $-n, \ldots, -1, 0, 1, \ldots, n$. Given a binary string $x \in \{0, 1\}^n$, we generate a star graph instance as

follows: First we make vertex $0$ the center and connect it to vertex $\pm 1$. For each $i \in \{1, \ldots, n-1\}$, if $x[i] = 0$ we connect vertex $i$ to $i+1$ and vertex $-i$ to $-i-1$. Otherwise, we connect $i$ to $-i-1$ and $-i$ to $i+1$. We illustrate this reduction in Figure 9.

The parity of $x$ can now be determined by the first step towards target vertex $n$. Specifically, if $\text{parity}(x) = 0$ (even), then the first vertex is $1$. Otherwise, it is $-1$. Consequently, if an algorithm can solve Task 1 for star graphs generated in this way, then it equivalently solves learning the full parity problem. $\qquad \square$

## B.2 Proof of Theorem 3

We prove this theorem by constructing a counterexample. Consider the distribution $D$ that is uniform over the sequences $\{ACA, BCB\}$. Our goal is to show that there exists an ideal next-token predictor, in the sense of Equation (5), that does not learn a belief state.

First, note that an ideal next-token predictor achieves a log loss of 1 bit on the first token and 0 log loss on all subsequent tokens. Define the forward encoder $\mathsf{F}(x_{1:t})$ to output a two-dimensional vector for any partial sequence $x_{1:t}$. Specifically, we define the encoding as follows:

$$\mathsf{F}(\emptyset) = (-1, -1),$$
$$\mathsf{F}(A) = \mathsf{F}(B) = (-1, 1),$$
$$\mathsf{F}(AC) = (1, -1),$$
$$\mathsf{F}(BC) = (1, 1).$$

Next, we define the output head $\mathsf{T}$ for the next-token predictions:

$$\mathsf{T}(-1, -1) = \text{uniform}(A, B),$$
$$\mathsf{T}(-1, 1) = C,$$
$$\mathsf{T}(1, -1) = A,$$
$$\mathsf{T}(1, 1) = B.$$

It can be verified that this setup achieves the optimal performance: a 1-bit log loss on the first token (since $A$ and $B$ are equally likely), and 0 log loss on subsequent tokens (since the continuation of the sequence is fully deterministic). However, the key observation is that $\mathsf{F}(A) = \mathsf{F}(B) = (-1, 1)$ is not a belief state for the remainder of the sequence. This is because any function $g(-1, 1)$ applied to this encoding outputs a distribution that is independent of whether the initial token was $A$ or $B$. Thus, the encoding fails to capture the information necessary to distinguish between the two possible continuations of the sequence, violating the definition of a belief state. Therefore, this next-token predictor does not output a belief state, completing the proof.

## B.3 Proof of Theorem 4

We again use a counterexample construction to prove this theorem. Consider the distribution $D$, which is uniform over the four sequences $\{DAA, DBB, SAB, SBA\}$. Here, the initial tokens $D$ and $S$ determine whether the next symbol is doubled or not. Our goal is to demonstrate that a multi-token model, in the sense of Equation (6), does not learn a belief state.

First, we construct an ideal multi-token model that suffers a log loss of 2 bits on each token except for the last token. Note that this is the minimal log loss achievable. Define the forward encoder $\mathsf{F}(x_{1:t})$ to output a two-dimensional vector for any partial sequence $x_{1:t}$. Specifically, we define the encoding as follows:

$$\mathsf{F}(\emptyset) = (-1, -1),$$
$$\mathsf{F}(D) = \mathsf{F}(S) = (-1, 1),$$
$$\mathsf{F}(DA) = \mathsf{F}(SB) = (1, -1),$$
$$\mathsf{F}(DB) = \mathsf{F}(SA) = (1, 1).$$

Now, define the output head for predicting the next tokens at different time steps as follows:

$$\mathsf{T}_1((-1,-1)) = \mathrm{uniform}(S, D),$$
$$\mathsf{T}_2((-1,-1)) = \mathrm{uniform}(A, B),$$
$$\mathsf{T}_3((-1,-1)) = \mathrm{uniform}(A, B),$$
$$\mathsf{T}_1((-1,1)) = \mathrm{uniform}(A, B),$$
$$\mathsf{T}_2((-1,1)) = \mathrm{uniform}(A, B),$$
$$\mathsf{T}_1((1,-1)) = A,$$
$$\mathsf{T}_1((1,1)) = B.$$

This multi-token model clearly achieves the minimal log loss. However, the key observation is that $\mathsf{F}(S) = \mathsf{F}(D) = (-1, 1)$ is not a belief state for the remainder of the sequence. The reason is that any function $g(-1, 1)$ applied to this encoding outputs a distribution that is independent of whether the initial token was $S$ or $D$. Thus, the encoding fails to capture the necessary information for predicting the future tokens. Therefore, this multi-token model does not output a belief state, completing the proof.

## C   STAR GRAPH DETAILS

In this section, we provide detailed descriptions of the star graph experiments from Section 3.4.

- **Data Generation.** We use the data generation code from the official codebase. To generate the star graph structure, each node $n_i$ is sampled uniformly from the set $\{1, \ldots, N\}$. For all experiments, we set $N = 50$, and we generate 8M examples for each data set.
- **Data Tokenization.** We follow the same tokenization settings as in (Bachmann & Nagarajan, 2024, Section G.1).
- **Architectures.** Both the forward and backward encoders consist of $n_{\mathrm{layers}} = 6$ layers with an embedding dimension of 768, $n_{\mathrm{head}} = 8$ attention heads, and an MLP expansion factor of 1. The baselines use the same configuration.
- **Training Details.** In all cases, we use the AdamW optimizer with a weight decay strength of 0.1. For $G(2, 5)$, the learning rate is set to $\eta = 3 \cdot 10^{-4}$, while for $G(5, 5)$ and $G(2, 20)$, a smaller learning rate of $\eta = 1 \cdot 10^{-4}$ is used. We run all experiments for 100 epochs to ensure convergence. All models run quickly, finishing 100 epochs in less than 2 hours. The belief state transformer reaches $\approx 100\%$ accuracy in a few minutes on the easier graphs. Each model is trained on a single A100 / H100 GPU with 80GB memory.
- **Evaluation.** We evaluate the models on 10,000 unseen graphs. To succeed, the model must output the entire path correctly, and we report the success rate.

### C.1   BASELINE DETAILS

We use the official codebase from Bachmann & Nagarajan (2024) to instantiate and train the baselines.

- **Forward-only.** This baseline trains a transformer to do next-token prediction over the sequence data. Teacher forcing is used during training, so the model gets ground truth intermediate sequences as input.
- **Data Augmentation.** Here, our data augmentation process entails simply modifying the data using some domain knowledge, before feeding it into the standard forward-only transformer training process.
  Given the original training sequences $[\mathcal{E} \mid n_1, n_l \mid n_1, n_2, n_3, \ldots n_l]$, we propose to replace the task specification $n_1, n_l$ with $n_1, n'$ where $n' \sim \{n_2 \ldots n_{l-1}\}$ is an intermediate node in the path. The resulting training sequence would then look like: $[\mathcal{E} \mid n_1, n' \mid n_1, n_2, n_3, \ldots n_l]$. The idea here is that replacing the long horizon goal with a subgoal closer to the start would make learning the goal conditioning easier.
- **Fill-in-the-Middle.** This is similar to the domain augmentation baseline, except we replace intermediate nodes with intermediate suffixes $n_j, \ldots n_T$. The input to the transformer is then: $[\mathcal{E} \mid n_1, n_T \mid n_j, n_{j+1}, \ldots n_l \mid n_1, n_2, n_3, \ldots n_l]$. During training time, we uniformly sample suffixes of different lengths.

- **Multi-token Prediction.** This baseline changes the objective from next-token prediction, to multiple token prediction. Given the graph description, start and goal, $[\mathcal{E} \mid n_1, n_l]$ the model needs to predict the path $[n_1, n_2, n_3, \ldots n_l]$ in a single forward pass.

## D    TINYSTORIES EXPERIMENTS

We train all models on a single A100 / H100 GPU with 80GB memory.

### D.1    TRAINING

The Belief State Transformer's encoders have the following settings: $n_{\text{layers}} = 8$, blocks with embedding dimension $e_{\text{dim}} = 768$, and $n_{\text{heads}} = 8$. The textheads $\mathsf{T}_n$, $\mathsf{T}_p$ are implemented as a single 3-layer MLP with dimensionality 512 and ReLU activations that outputs two predictions. The total model size is 80 million parameters.

The FIM baseline trains a forward only transformer with the following settings: $n_{\text{layers}} = 12$, blocks with embedding dimension $e_{\text{dim}} = 768$, and $n_{\text{heads}} = 8$. The total model size is 85 million parameters.

The FIM baseline's training is simple—for a given prefix $x_{1:t}$, we prepend a random suffix $x_{k:T}$ for $k \sim \mathcal{U}(t+2, T)$ and train the transformer to predict $x_{t+1}$. We create as many prefix-suffix inputs as the GPU memory allows. Even then, for longer sequences, it becomes increasingly intractable for the FIM baseline to train on all possible prefix-suffix inputs.

The Belief State Transformer on the other hand, is able to train on all $O(T^2)$ possible prefix-suffix combinations for a sequence of length $T$, by carefully accumulating text head gradients over all pairs before computing the forward and backward encoder gradients. Without such a scheme, training goes from a few hours to multiple days. As a result, the Belief State Transformer is only around $3 - 5\times$ slower than the FIM baseline's suffix-sampling update, yet is able to train on all $O(T^2)$ possible prefix-suffix inputs.

We trained the Belief State transformer on 1 epoch of the TinyStories dataset with 2.7M stories, with a batch size of 256. The training takes around 5 hours for 1 epoch. Because the FIM baseline goes through an epoch more quickly than the Belief State transformer, we allow it to run for multiple epochs with a max training time of 5 hours, and use early stopping on the validation loss to select the best checkpoint.

---

**Algorithm 2** Beam Search

---

**Require:**  text $x_{1:t}$, goal $x_{t+k:T}$, number of steps $K$, beams $n$
1:  Priority Queue $Q_0 \leftarrow (1, \emptyset)$
2:  **for** $k = 1$ to $K - 1$ **do**
3:      **for** $j = 1$ to $n$ **do**
4:          **if** $|Q_{k-1}| > 0$ **then**
5:              (priority $r$, sequence $u$) $\leftarrow \text{pop}(Q_{k-1})$
6:              **for** Possible $\tilde{x}$ **do**
7:                  $Q_k \leftarrow (r\mathsf{T}_n(\tilde{x} \mid \mathsf{F}(x_{t+k:T} + x_{1:t} + u)), u + \tilde{x})$
8:  **Output:** $\text{top}(n, Q_K)$

---

The FIM model uses beam search (Graves, 2012) to find high probability sequences at inference time. At a high level, beam search only examines and expands upon a fixed amount of paths during search to keep search tractable. To keep the inference time resources comparable, We use the same number of paths as trajectories we generate in Algorithm 1, which is 120.

### D.2    EVALUATION

We evaluate the models on 100 held out stories from the TinyStories dataset. For each story, we use the first 50 tokens for the prompt, and last 100 tokens for the suffix. The choice of 100 for the suffix length is due to the varying length of the stories - shorter stories would often just receive padding tokens as the suffix if the suffix window wasn't long enough.

We use GPT4 to judge the quality of generated text. Given two generated stories, one from each model, we ask GPT4 to analyze the stories and output a recommendation on which story is superior. The stories are anonymized and shuffled to prevent information leakage about its author before we feed it to GPT4. For every evaluation, we shuffle the story ordering once, and repeat the the evaluation 3 times for a total of 6 GPT4 evaluations for every story. Since we evaluate across 100 held out stories, this amounts to 600 total queries to GPT4. We use the OpenAI API to automate the process.

Here is an example input we feed into GPT4.

**System Prompt:** *You are evaluating two story generation models. First, analyze the outputs and note their properties. Then, make an argument for why one is better than another, or say that both are roughly equal.*

Story 1:

Once upon a time, in a warm and sunny place, there was a big pit. A little boy named Tom liked to play near the pit. One day, Tom lost his red ball. He was very sad. Tom asked his friend, Sam, "Did you see my red ball?" Sam said, "No, but let's look together." They looked and looked, but they could not find the red ball. Tom was very sad. Then, a big bird came and dropped the red ball. The bird had the red ball in its beak. Tom and Sam were so happy! They said, "Thank you, bird!" The bird flew away, and Tom played with his red ball all day.

Story 2:

Once upon a time, in a warm and sunny place, there was a big pit. A little boy named Tom liked to play near the pit. One day, Tom lost his red ball. He was very sad. Tom asked his friend, Sam, for help. "Sam, can you help me find my red ball?" Sam said, "Yes, I will help you." They looked and looked, but they could not find the red ball. Just when they were about to give up, a big bird flew down from the sky. The bird had the red ball in its beak! The bird dropped the ball into the pit. Tom and Sam were so happy. They thanked the bird and played with the red ball

Then, GPT4 responds with:

**Analysis of Outputs:**

Story 1:

**Grammar and Style:** - Simple and clear sentences, with correct grammar throughout. - Repetitive usage of "Tom was very sad". - "Tom and Sam were so happy!" creates a clear and optimistic conclusion.

**Narrative Flow:** - Introduction: Sets the scene with Tom and the pit. - Conflict: Tom loses his red ball. - Resolution: A big bird returns the ball. - Conclusion: Ends with Tom playing all day, providing a satisfying closure.

Story 2:

**Grammar and Style:** - Generally grammatically correct. - Uses a more varied sentence structure. - Dialogue includes direct speech ("Sam, can you help me find my red ball?"), adding a personal touch.

**Narrative Flow:** - Introduction: Similar setup with Tom and the pit. - Conflict: Tom loses his red ball, seeks Sam's help. - Resolution: Big bird returns the ball, but drops it into the pit. - Conclusion: The story seems to end abruptly without further explanation about retrieving the ball from the pit.

Argument for Which Story is Better:

**Conclusion Analysis:** - **Story 1** concludes clearly with Tom playing all day, which establishes a definite and satisfying ending for the reader. - **Story 2** leaves a gap in logic—though the bird returns the ball, it drops it into the pit, and we don't see how Tom and Sam retrieve it. The absence of this information feels like an incomplete ending.

**Grammar and Narrative:** - Both stories are grammatically correct and maintain simple language, making them suitable for young readers. - Story 1 employs more repetition, which could be beneficial for reinforcement but potentially tedious for some readers. Story 2 introduces more varied dialogue which might be more engaging.

**Overall Conclusion:** - Given that Story 1 has a logical setup, development, and conclusion, **Story 1** is the better option. It provides a clear, satisfying end where Tom is happy and continues to play. Story 2, while containing more engaging dialogue, fails to wrap up the narrative logically, leaving the reader with lingering questions about the outcome.

As seen in the example above, GPT4 is able to consider multiple factors about the story, from low level grammar and vocabulary choices, to higher level concepts like cohesiveness and logical flow, and uses these factors to make its final recommendation.

### D.3 MODEL OUTPUTS

Here are additional stories generated by the Belief State transformer in the goal-conditioned setting.

**Story 1:** You don't have to be scared of the loud dog, I'll protect you". The mole felt so safe with the little girl. She was very kind and the mole soon came to trust her. He was so gentle and kind. The little girl was so happy to have a new friend. She gave the mole a big hug and they played together all day. They ran around the garden and laughed and had lots of fun. When it was time to go home, the little girl hugged the mole and said goodbye. The mole was so happy to have a friend like her and they promised to always be friends.

**Story 2:** Once upon a time, in a warm and sunny place, there was a big pit. A little boy named Tom liked to play near the pit. One day, Tom lost his red ball. He was very sad. Tom asked his friend, Sam, "Did you see my red ball?" Sam said, "No, but let's look together." They looked near the pit and found the red ball. Tom was very happy. He said, "Thank you, Sam!" But then, a big dog came and took the red ball. Tom and Sam were scared. They wanted the ball back. They ran after the dog. The dog ran into the pit and disappeared. Tom and Sam were sad. They went into the pit to search. It was dark and scary. They could not find the ball. They tried to get out, but the pit was too deep. Tom and

Sam were stuck in the pit. They called for help, but no one could hear them. They were sad and scared, and they never got out of the pit.

**Story 3:** Once upon a time there was a little girl named Lucy. She loved to go to the store to buy sweets with her mom and dad. On this special day, Lucy entered the store with her mom. Lucy saw a big, red ball. She asked her mom, "Can I have the ball, please?" Her mom said, "No, Lucy. We don't have enough money today. We have to go home and get a new one." Lucy was sad, but she understood. She asked her mom, "Can we get a new ball?" Her mom smiled and said, "Yes, we can get a new one. But first, we have to be peaceful and kind to her. Have patience and understanding. Together, you can both be happy!" So, Lucy smiled at the girl and said, "Can we play together?" The little girl softened and smiled back. She agreed to share the toy and even let Lucy have a turn first. Lucy and the little girl played together happily.

And some outputs for the FIM baseline, over the same prefix / suffix pairs from above.

**Story 1:** You don't have to be scared of the loud dog, I'll protect you". The mole felt so safe with the little girl. She was very kind and the mole soon came to trust her. He said, "Don't be scared, I'm here to protect you". The little girl was so happy and thanked the mole for keeping her safe. From then on, the little girl and the mole were the best of friends.

**Story 2:** Once upon a time, in a warm and sunny place, there was a big pit. A little boy named Tom liked to play near the pit. One day, Tom lost his red ball. He was very sad. Tom asked his friend, Sam, for help. "Sam, can you help me find my red ball?" Tom said. Sam said, "Yes, I will help you." They looked and looked, but they could not find the red ball. They were very sad. Then, a big bird flew down from the sky. The bird had the red ball in its beak! The bird dropped the ball into the pit. Tom and Sam were so happy. They said, "Thank you, bird!"They went into the pit to search. It was dark and scary. They could not find the ball. They tried to get out, but the pit was too deep. Tom and Sam were stuck in the pit. They called for help, but no one could hear them. They were sad and scared, and they never got out of the pit.

**Story 3:** Once upon a time there was a little girl named Lucy. She loved to go to the store to buy sweets with her mom and dad. On this special day, Lucy entered the store with her mom and dad. When they arrived at the store, Lucy was so excited to see all the different kinds of sweets. She couldn't believe her eyes when she saw all the different kinds of sweets. She couldn't believe her eyes when she saw all the different kinds of sweets. Lucy's mom smiled at her and said, " to be peaceful and kind to her. Have patience and understanding. Together, you can both be happy!" So, Lucy smiled at the girl and said, "Can we play together?" The little girl softened and smiled back. She agreed to share the toy and even let Lucy have a turn first. Lucy and the little girl played together happily.

# E  TRAINING THE BELIEF STATE TRANSFORMER

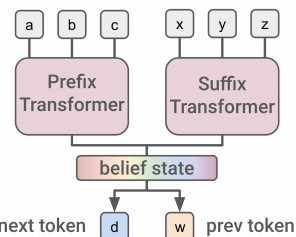

Figure 10: BST input and outputs.

The belief state transformer, at a high level, takes in a prefix $x_{1:i}$, a suffix $x_{i+k:T}$, and predicts the token right after the prefix, and the token right before the suffix. See Figure 10 for an example.

To create training data from a sequence $x_{1:T}$, all valid prefix-suffix pairs are created from the sequence. We define a valid prefix-suffix pair as follows.

Let the prefix be $x_{1:i}$. Then a suffix $x_{i+k:T}$ is valid for all $k \geq 2$. This condition ensures that there is at least one token that separates the prefix and suffix, which makes it feasible to predict tokens in between the prefix and suffix.

For a sequence $n$ tokens long, there are $O(n^2)$ possible valid prefix-suffix pairs. See Figure 11 for example training inputs and their targets.

We present pytorch pseudocode illustrating a simplified implementation of the belief state transformer (Figure 12). Additionally, we present a slightly more complex version which computes and accumulates the gradients for the text head MLP before backpropagating gradients into the encoders (Figure 13). This implementation is more computationally efficient as it avoids backpropagating through the large transformer encoders multiple times. Gloeckle et al. (2024) adopts a similar strategy to efficiently compute gradients of an encoder with multiple heads.

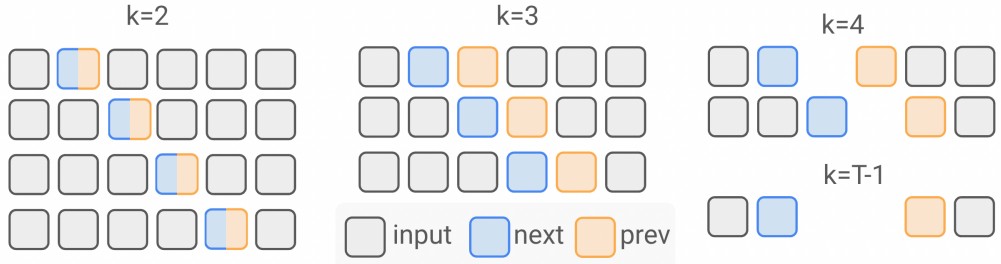

Figure 11: Forming training examples for the BST.

We define the training overheads. Let the sequence length be $n$. To scale with batch size, one would multiply the batch size with the following equations. Let $h$ be the cost related to the output head size, and $e$ be the cost of soft attention.

GPT: $O(hn+en^2)$, first term is cost of doing `text_head`$(f_i)$ for $f_{1:t}$, second term is cost of encoding the tokens $x_{1:t}$ with attention to produce $f_{1:t}$.

BST: $O(hn^2 + en^2)$, first term is cost of doing `text_head`$(f_i, b_j)$ for all valid pairs in the $O(n^2)$ prefix-suffix pairs, second term is cost of encoding the prefix and suffix tokens with attention. Note that at inference, BST is $O(hn + en^2)$.

```python
import torch
import torch.nn as nn

def belief_state_objective(enc_F, enc_B, text_head, x):
  bs, T = x.shape
  forward_state = enc_F(x)
  backward_state = enc_B(x.flip(1)).flip(1)
  ft = torch.arange(T, dtype=torch.int32)
  bt = torch.arange(T, dtype=torch.int32)
  combinations = torch.cartesian_prod(ft, bt)
  combinations = combinations[(combinations[:, 1]-combinations[:, 0]>=2)]
  fb_pairs = combinations.clone()
  fb_pairs = fb_pairs[combinations[:,1] < T]
  f_idxs = fb_pairs[:, 0]
  b_idxs = fb_pairs[:, 1]
  nt_idxs = (combinations[:, 0] + 1)
  f = forward_state[:, f_idxs]
  b = backward_state[:, b_idxs]
  single_labels_f = x[:, nt_idxs].unsqueeze(2)
  single_labels_b = x[:, b_idxs].unsqueeze(2)
  single_labels = torch.cat((single_labels_f, single_labels_b), dim=2)
  logits = text_head(torch.cat([f, b],dim=2))
  fb_numpairs = fb_pairs.shape[0]
  logits = logits.reshape((bs, fb_numpairs, 2, -1))
  logits = logits.reshape((bs*fb_numpairs*2, -1))
  single_labels = single_labels.reshape((bs*fb_numpairs*2))
  loss = nn.CrossEntropyLoss()(logits, single_labels)
  return loss

if __name__ == '__main__':
  batch_size = 8
  T = 12
  m = 512
  num_tokens = 100
  #Use dummy function in place of actual autoregressive transformer
  enc_F = nn.Sequential(nn.Embedding(num_tokens, m), nn.Linear(m, m))
  enc_B = nn.Sequential(nn.Embedding(num_tokens, m), nn.Linear(m, m))
  text_head = nn.Sequential(nn.Linear(m*2, m), nn.LeakyReLU(), nn.Linear(
                                        m, num_tokens*2))
  x = torch.randint(0, num_tokens, size=(batch_size,T))

  loss = belief_state_objective(enc_F, enc_B, text_head, x)
  print(loss)
```

Figure 12: A simple implementation of the belief state transformer objective

```python
import torch
import torch.nn as nn

def belief_state_objective(all_f, all_b, text_head, x):
  bs, T = x.shape
  forward_state = all_f
  backward_state = all_b.flip(1)
  ft = torch.arange(T, dtype=torch.int32)
  bt = torch.arange(T, dtype=torch.int32)
  combinations = torch.cartesian_prod(ft, bt)
  combinations = combinations[(combinations[:, 1]-combinations[:, 0]>=2)]
  fb_pairs = combinations.clone()
  fb_pairs = fb_pairs[combinations[:,1] < T]
  f_idxs = fb_pairs[:, 0]
  b_idxs = fb_pairs[:, 1]
  nt_idxs = (combinations[:, 0] + 1)
  f = forward_state[:, f_idxs]
  b = backward_state[:, b_idxs]
  single_labels_f = x[:, nt_idxs].unsqueeze(2)
  single_labels_b = x[:, b_idxs].unsqueeze(2)
  single_labels = torch.cat((single_labels_f, single_labels_b), dim=2)
  logits = text_head(torch.cat([f, b],dim=2))
  fb_numpairs = fb_pairs.shape[0]
  logits = logits.reshape((bs, fb_numpairs, 2, -1))
  logits = logits.reshape((bs*fb_numpairs*2, -1))
  single_labels = single_labels.reshape((bs*fb_numpairs*2))
  loss = nn.CrossEntropyLoss()(logits, single_labels)
  return loss

if __name__ == '__main__':
  batch_size = 8
  T = 12
  m = 512
  num_tokens = 100
  #Use dummy function in place of actual autoregressive transformer
  enc_F = nn.Sequential(nn.Embedding(num_tokens, m), nn.Linear(m, m))
  enc_B = nn.Sequential(nn.Embedding(num_tokens, m), nn.Linear(m, m))
  text_head = nn.Sequential(nn.Linear(m*2, m), nn.LeakyReLU(), nn.Linear(
                                      m, num_tokens*2))
  x = torch.randint(0, num_tokens, size=(batch_size,T))
  f = enc_F(x)
  b = enc_B(x)

  # just get the text head computation graph
  _f = f.detach()
  _b = b.detach()
  _f.requires_grad = True
  _b.requires_grad = True

  loss = belief_state_objective(_f, _b, text_head, x)
  # compute text head gradients over all prefix/suffix pairs.
  loss.backward()
  # Update encoders with 1 backward pass.
  f.backward(_f.grad)
  b.backward(_b.grad)
```

Figure 13: Efficient computation of all prefix-suffix losses.

