# OpenReview forum: "The Belief State Transformer"
_ICLR.cc/2025/Conference — ICLR 2025 Poster_

### Official Review · Reviewer_1PYd · 2024-10-29

**Soundness:** 3
**Presentation:** 3
**Contribution:** 3
**Rating:** 6
**Confidence:** 3

**Summary:**

The paper introduces the Belief State Transformer, a next-token predictor that leverages both prefix and suffix inputs to enhance prediction capabilities. By learning a compact belief state, the model overcomes the limitations of traditional forward-only transformers, excelling in goal-conditioned tasks such as story writing.

**Strengths:**

- The idea of learning a compact belief state that captures all relevant information for future predictions is pretty interesting.
- The paper is mostly well written, with controlled experiments and analysis of the proposed method.

**Weaknesses:**

- The methods and implementations are somewhat confusing, as detailed in the question section.
- There is no analysis regarding scaling AR and the Belief State Transformer. Considering cost issues, it might be possible to examine this on relatively small series models. The concern is whether the forward AR inherently encodes a "belief state" as AR model size increases.

**Questions:**

- Based on the model parameter details in section C.1, are the forward and backward encoders shared? If so, does this mean that by simultaneously learning the backward objective, the forward encoder obtains some future information, leading to a better representation of $f_t$? Furthermore, would not using a shared encoder cause the model to fail?
- I'm a bit confused about the proof about Theorem 3. Given the proof of Theorem 2, it seems we can also derive a similar proof to show AR also encodes a belief state through forward decomposition. The authors provide a counterexample for AR, but stating that "F(A)=F(B)=(−1,1)” seems unreasonable, as a normal encoder won't produce exactly the same encodings for different inputs.
- Could the authors elaborate more on lines 725-726 and show how the belief state transformer constructs a belief state that corresponds to the {ACA, BCB} distribution?
- Why the belief state transformer performs better than FIM training? Could the authors give some intuitions about this?

---

> ### Author Response · Authors · 2024-11-16
> **(11/16) Response**
>
> Thanks for the review!
>
> >There is no analysis regarding scaling AR and the Belief State Transformer. Considering cost issues, it might be possible to examine this on relatively small series models. The concern is whether the forward AR inherently encodes a "belief state" as AR model size increases.
>
> This is a real concern, but it appears to require significant scale in both data and parameters to test this concern since small-scale solutions have been ruled out in the star graph paper [1].
>
> [1] Bachmann, Gregor, and Vaishnavh Nagarajan. "The pitfalls of next-token prediction." ICML (2024).
>
> > Based on the model parameter details in section C.1, are the forward and backward encoders shared? If so, does this mean that by simultaneously learning the backward objective, the forward encoder obtains some future information, leading to a better representation of f_t? Furthermore, would not using a shared encoder cause the model to fail?
>
> No, they are not shared. Note that the BST uses 2 encoders with 8 layers, and the combined parameter count of these 2 shallower encoders is roughly the same as the FIM’s 12-layer encoder.
>
> > I'm a bit confused about the proof about Theorem 3. Given the proof of Theorem 2, it seems we can also derive a similar proof to show AR also encodes a belief state through forward decomposition. The authors provide a counterexample for AR, but stating that "F(A)=F(B)=(−1,1)” seems unreasonable, as a normal encoder won't produce exactly the same encodings for different inputs.
>
> We agree that empirically F(A) and F(B) might sometimes have different representations.  However, that is not sufficient to establish that a compact belief state is learned since the proof in theorem 2 applies always, not merely "sometimes".
> The normal autoregressive model (GPT) does not learn a compact belief state through forward decomposition, because there is a dependence on all previous tokens via attention across time, with no explicit dependence on a belief state.
>
> If you want another intuitive counterexample for GPT, there’s an old toy problem called copying where a sequence is modeled [x1 x2 x3 x4 0 0 0 0 0 x1 x2 x3 x4], i.e. the model is tasked with copying a sequence.  This is fairly easy for transformers, and the true belief state at the midpoint should have enough information to reconstruct [x1 x2 x3 x4].  However, a GPT-style transformer does not need to express any information about the sequence on the midpoint step’s final layer representation, because it only needs to predict the padding token on that step.
>
> > Could the authors elaborate more on lines 725-726 and show how the belief state transformer constructs a belief state that corresponds to the {ACA, BCB} distribution?
>
> The key observation is that F(A)=F(B) does not optimize the Belief State Transformer objective since T_p(F(A), B(∅)) is optimized to predict A and T_p(F(B),B(∅)) is optimized to predict B.
>
> > Why the belief state transformer performs better than FIM training? Could the authors give some intuitions about this?
>
> There are two important effects here: (a) The Belief State Transformer is trained on O(n^2) gradients whereas FIM is trained on only O(n) gradients per update.  Additionally the gradients include both the next and prev predictions, which lead to learning a belief state and could also be a richer training signal. (b) The BST next head can provide an implicit valuation of generation by calculating the probability of goal tokens given the generation.  In the FIM paradigm, this is not possible because the “middle” comes after the suffix.

---

> ### Author Response · Authors · 2024-11-26
> **(11/26/24) Followup**
>
> Dear reviewer 1PYd,
>
> As the discussion period draws to a close, we’d like to check back to see if you have any remaining concerns. We believe we sufficiently answered your questions, but would be happy to clarify further. We would also be grateful for any additional comments or feedback.

---

> > ### Author Response · Authors · 2024-12-02
> > **(12/2/24) Last day for us to respond**
> >
> > Hello reviewer 1PYd, we'd like to check in to see if there are any remaining questions or concerns before the author reply window closes.

---

### Official Review · Reviewer_Aowt · 2024-10-31

**Soundness:** 3
**Presentation:** 4
**Contribution:** 3
**Rating:** 8
**Confidence:** 4

**Summary:**

This work presents a new Transformer model architecture capable of both next and previous token prediction. The model leverages two separate auto-regressive Transformer decoders, one processing input sequences forward and the other backward, the outputs of which are concatenated together and fed into the model’s output head to predict each token conditional on its prefix and suffix.  Apart from efficient training, this design enables the model, called Belief State Transformers (BSTs), to effectively learn good representations for token prediction, supports more flexible inference (such as goal-conditioned decoding or backward evaluation),  and demonstrates success on complex tasks, such as star graphs, where conventional next-token-prediction Transformers are known to fall short.  In addition, BSTs also achieve better performance in story-infilling tasks than prior baselines.

**Strengths:**

- This paper addresses an important problem in improving the non-myopic prediction capabilities of current auto-regressive language models. Although this study is limited to small-scale settings, its design contributes meaningfully to expanding the architectural space of flexible language models.
- The proofs in Section 4 further strengthen the findings and nicely highlight the biases present in modeling non-myopic dependencies across various objectives (namely BSTs, next token prediction, and teacher-less training).
- The probing experiments, as demonstrated in Figure 5, Section 5.4, are particularly interesting, as they effectively showcase the improved quality of learned bidirectional representations.

**Weaknesses:**

The study lacks experiments on more realistic, large-scale settings or with larger model sizes. For instance, evaluating bidirectional representations in tasks like code infilling could provide more robust insights.

**Questions:**

1. In Line 269, the footnote mentions “the fact that the forward-only approach does not produce a compact belief state is well known”. Could you clarify what defines a belief state as “compact”?
2. How does FIM perform specifically on star graph tasks?
3. What is the training overhead of BSTs compared to conventional auto-regressive Transformer decoders, particularly in terms of sequence length and batch sizes?

---

> ### Author Response · Authors · 2024-11-16
> **(11/16/24) Response**
>
> Thanks for the review!
> > The study lacks experiments on more realistic, large-scale settings or with larger model sizes. For instance, evaluating bidirectional representations in tasks like code infilling could provide more robust insights.
>
> We look forward to scaling and investigating harder domains in future work. Code infilling is a natural task that we are investigating.
>
> > In Line 269, the footnote mentions “the fact that the forward-only approach does not produce a compact belief state is well known”. Could you clarify what defines a belief state as “compact”?
>
> We refer to the embedding input to the text head, as the compact belief state. It is desirable to capture the belief state into a low-dimensional representation like the text head embedding input, as this makes it ergonomic for downstream tasks.
>
> > How does FIM perform specifically on star graph tasks?
>
> Multiple reviewers have asked about FIM on the Stargraph setting; we will look into running FIM on the Stargraph and aim to report results before the end of the discussion period.
>
>
> > What is the training overhead of BSTs compared to conventional auto-regressive Transformer decoders, particularly in terms of sequence length and batch sizes?
>
> Here are the training overheads in terms of the sequence length $n$. To scale with batch size, one would multiply the batch size with the following equations. Let $h$ be the cost related to the output head size, and $e$ be the cost of soft attention.
>
> GPT:
> $O(h n+e n^2)$, first term is cost of doing text\_head$(f_i)$ for $f_{1:n}$, second term is cost of encoding the tokens $x_{1:n}$ with attention to produce $f_{1:n}$.
>
> BST:
> $O(h n^2 + e n^2)$, first term is cost of doing text\_head$(f_i,b_j)$ for all valid pairs in the $O(n^2)$ prefix-suffix pairs, second term is cost of encoding the prefix and suffix tokens with attention.
> Note that at inference, BST is $O(h n + e n^2)$.

---

> > ### Comment · Reviewer_Aowt · 2024-11-24
> >
> > I thank the authors for their comprehensive responses and their efforts to address my concerns. After reading the responses and the other reviews, I have decided to maintain my original score, as it reflects my assessment of the paper's contributions. I look forward to the forthcoming empirical results on FIM in star graph tasks.

---

> ### Author Response · Authors · 2024-11-26
> **(11/26/24) Response**
>
> Dear reviewer Aowt,
> Thank you for the response. We have some updates to report:
>
> Per your suggestion, we ran the FIM baseline on the Stargraph, and found that it performs poorly, picking random paths. We believe this is due to the next token prediction objective of FIM, which is known to fail on Stargraph. We added these findings to the Stargraph experimental section.
>
> We also incorporated the discussion on training overhead to the appendix, which will be helpful for future readers.
>
> We would be grateful for any additional comments or feedback before the discussion period closes.

---

> > ### Author Response · Authors · 2024-12-02
> > **(12/2/24) Last day for us to respond**
> >
> > Hello reviewer Aowt, we'd like to check in to see if there are any remaining questions or concerns before the author reply window closes.

---

### Official Review · Reviewer_Yi5e · 2024-11-03

**Soundness:** 2
**Presentation:** 2
**Contribution:** 3
**Rating:** 6
**Confidence:** 4

**Summary:**

This paper introduces the Belief States Transformer, a model that predicts the next token for the left context (prefix) and the previous token for the right context (suffix) using two distinct encoders. The authors evaluate the proposed approach on two tasks: finding paths on star-shaped graphs and generating stories using the TinyStories dataset. The results show that the proposed method outperforms models trained only in the left-to-right direction and the Fill-in-the-Middle (FIM) approach, which predicts the next token conditioned on both left and right contexts.

**Strengths:**

- The paper addresses the important challenge that modern architectures lack planning ahead mechanisms to rely on before generating solutions.
- By predicting both the next token for the left context and the previous token for the right context, the model uses a more complex training scenario with more training signal (O(n^2)) compared to FIM and regular next-token language modeling.
- Despite the more complex training, the model retains the same autoregressive inference mechanism as regular language models, ensuring compatibility with existing model inference scenarios.
- The authors select a set of tasks (path finding on star-shape graphs and story generation) that test models' abilities to make generations that follow certain goals. They show that proposed Belief State Transformers outperform regular next-token prediction models and FIM, indicating the effectiveness of their approach.

**Weaknesses:**

- Comparison with FIM on Star-Shaped Graphs. The paper does not provide results for the Fill-in-the-Middle (FIM) approach on the star-shaped graph task. Including FIM as a baseline in this task would offer a more comprehensive comparison and help determine whether the improvements are due to the model architecture or the specific training objectives.
- The paper does not discuss whether a backward language model could solve the star-shaped graph task more effectively than a forward LM. Exploring this could provide insights into the benefits of using bidirectional information in the model.
- Models like BERT, trained with a Masked Language Modeling (MLM) objective, consider information from both directions and can be adapted for generation with additional techniques (e.g., refer to “BERT has a Mouth, and It Must Speak: BERT as a Markov Random Field Language Model”) or generate multiple tokens at once. The paper does not compare Belief State Transformer with MLM models, which could highlight the advantages or limitations of the proposed approach.
- Since the star-shaped graph task is stated to be equivalent to parity problems, the authors do not demonstrate the model’s ability to solve parity tasks. Including such experiments would strengthen the claim about the model’s capabilities in handling these types of problems and prove that the proposed approach is superior to regular next-token transformers.
- The paper lacks clarity in description of some results and experiments (refer to questions section).

**Questions:**

- It is unclear how the model handles unconditional text generation. Specifically, does the generation proceed left-to-right with only the scoring head for the reverse direction, or some other way?
- The term "teacherless" (Fig. 2) might be confusing. Would it be more appropriate to refer to this approach as "multi-token prediction"? Clarifying the terminology could improve the paper’s readability.
- Figure 5, left: It is not clear whether the probe tests the model's ability to predict nodes on the correct path several steps ahead, or nodes that the model would generate an inference (including possible errors). Forward-only model has low scores on star graph tasks, but could it possibly predict its own outputs based on internal state?
- Figure 6: What is meant by “graph description”? What is on the x-axis? It is challenging to interpret the results presented in Figure 6.
- On TinyStories authors use goal-conditioned planning algorithm for inference (Alg.1). Can this algorithm be adapted to FIM?
- The paper does not specify whether FIM was trained on full sequences or if samples were truncated during training. Was FIM trained on full sequences or truncated samples? Could training on incomplete sequences or limitations in generation length impact its ability to generate complete outputs? Could FIM and BST results from Sec 5.2 be improved by setting suffix to “.”, so both models would know that they should end generation with end of sentence.
- There are several aspects that should be made clear for each experiment: is generation done from left-to-right or right-to-left? Are generations scored based on the next-token head, previous-token head, or both?

Related work could mention BERT as a bidirectional model trained with MLM objective and ELMo as one of the first popular language models that incorporated information from two directions.

---

> ### Author Response · Authors · 2024-11-16
> **(11/16/24) Response 1/2**
>
> Thank you for your review!
>
> > Is generation done from left-to-right or right-to-left? Are generations scored based on the next-token head, previous-token head, or both?
>
> As mentioned in Section 2.2 Belief State Inference, we always do left-to-right generation. When doing conditional planning, we score based on the next token head, and when doing unconditional, we use the previous token head.
>
> > Including FIM as a baseline in this task would offer a more comprehensive comparison and help determine whether the improvements are due to the model architecture or the specific training objectives.
>
> It seems like the reviewer would like to know where the performance improvements of the BST are coming from for the Stargraph task, and the reviewer thinks a FIM baseline would give more information on the performance gains of the BST.
>
> However, this is already answered by the experiments. The Belief State Transformer uses standard left-to-right autoregressive generation during inference time, identical to the standard next token predictor models, and outperforms the standard next token predictor model. Hence, all the change in performance is due to the change in the objective, which is detailed in the ablation section.
>
> On a related note, Reviewer Aowt also questioned how well FIM would do in the Stargraph setting. Because multiple reviewers have asked about FIM on the Stargraph setting, we will look into running FIM on the Stargraphs and aim to report results before the end of the discussion period.
>
> > Does not discuss whether a backward language model could solve the star-shaped graph task more effectively than a forward LM.
>
> A language model that processes sequences from right to left could solve the Stargraph, and was in fact proposed by the Stargraph paper as a solution. However, such a solution is very task-specific and requires modification of both input data and model outputs (specifically, the user must reverse the training sequences and model outputs).
>
> As a result, we limited our comparisons to more general baselines like teacherless (multi-token prediction), and data augmentation. The reviewer may be interested in [1] which shows that LLMs trained to predict right-to-left are generally worse than LLMs trained left-to-right in natural language tasks, which provides further evidence on the limitation of backward models.
> We will update the text to make it clear that the Stargraph can be easily solved using domain knowledge and that our motivation is to develop a general model that can solve hard tasks like Stargraph without domain knowledge.
>
> [1] Papadopoulos, Vassilis, Jérémie Wenger, and Clément Hongler. "Arrows of Time for Large Language Models." arXiv preprint arXiv:2401.17505 (2024).
>
> > Models like BERT, trained with a Masked Language Modeling (MLM) objective, consider information from both direction… The paper does not compare Belief State Transformer with MLM models …Related work could mention BERT as a bidirectional model trained with MLM objective and ELMo as one of the first popular language models that incorporated information from two directions.
>
> Thanks for the suggestion, and we will update related work.  Non-causal models like BERT could ostensibly handle the Stargraph dataset better than GPT-style transformers. However, as motivated in our intro, we are interested in examining the flaws of the dominant sequence modeling paradigm (GPT + next token prediction), and addressing them.
>
> Our contribution, the Belief State Transformer, is a generalization of the GPT architecture, and inherits key desirable properties (i.e. causal attention, left to right decoding) while overcoming existing weaknesses in GPT. Thus, our main point of comparison is against prior GPT-style approaches.
>
> > Since the star-shaped graph task is stated to be equivalent to parity problems, the authors do not demonstrate the model’s ability to solve parity tasks.
>
> Whether or not BSTs can solve parity isn’t an interesting question.  Our primary objective with the Stargraph experiment, is to present a simple and well-known problem in which GPT models empirically fail, and show that BST outperforms the prior work. The reduction between the Stargraph task with the parity problem is to give intuition behind why the Stargraph is hard to solve for next token predictors.

---

> > ### Author Response · Authors · 2024-11-16
> > **(11/16/24) Response 2/2**
> >
> > > The term "teacherless" (Fig. 2) might be confusing. Would it be more appropriate to refer to this approach as "multi-token prediction"? Clarifying the terminology could improve the paper’s readability.
> >
> > You’re right, but we use “teacherless” to be consistent with the terminology in the Stargraph paper. Changing this would make it harder to follow the literature.
> >
> > > Figure 5, left: It is not clear whether the probe tests the model's ability to predict nodes on the correct path several steps ahead, or nodes that the model would generate an inference (including possible errors). Forward-only model has low scores on star graph tasks, but could it possibly predict its own outputs based on internal state?
> >
> > The former -  probe tests the model’s ability to predict nodes on the correct path several steps ahead. We will update the wording to clarify this.  It’s not completely obvious if the forward-only model could predict future tokens in a sampled sequence, because these may depend on what tokens are sampled by the model on future steps.
> >
> > > Figure 6: What is meant by “graph description”? What is on the x-axis? It is challenging to interpret the results presented in Figure 6.
> >
> > The graph description is the edge list. The x-axis is the index of each element in the graph description. The point of figure 6 is to show that the belief state representation contains more information about the structure of the graph than the next token representation.
> >
> > > On TinyStories authors use goal-conditioned planning algorithm for inference (Alg.1). Can this algorithm be adapted to FIM?
> >
> > Yes, Alg 1. could be adapted for FIM, but is not straightforward. For example, the scoring function uses the BST’s previous head in the unconditional setting.
> >
> > > The paper does not specify whether FIM was trained on full sequences or if samples were truncated during training. Was FIM trained on full sequences or truncated samples? Could training on incomplete sequences or limitations in generation length impact its ability to generate complete outputs? Could FIM and BST results from Sec 5.2 be improved by setting suffix to “.”, so both models would know that they should end generation with end of sentence.
> >
> > All models are trained on full sequences, so the models should learn to complete outputs. We actually tried setting the suffix to  “.” in an early experiment, but didn’t find any noticeable improvements over using the empty suffix.

---

> ### Author Response · Authors · 2024-11-26
> **(11/26/24) Response**
>
> Dear reviewer Yi5e,
> Thank you again for your detailed and constructive comments. We’ve updated the draft and ran some more experiments, and would like to report in.
>
> Per your suggestion, we ran the FIM baseline on the Stargraph, and found that it performs poorly, picking random paths. We believe this is due to the next token prediction objective of FIM, which is known to fail on Stargraph. We added these findings to figure 2 and section 3.4.
>
> We’ve updated the text to emphasize that we use left-to-right ordering in all experiments, and that we use the next head for the conditional experiments, and previous head for unconditional experiments. See lines 116, 354, and 416.
>
> Following your suggestion, we’ve updated the related work to mention non-causal approaches like ELMo and BERT, see Section 6.
>
> We believe that we’ve sufficiently answered all of your questions, but would be happy to engage further, and would be grateful for any other feedback.

---

> > ### Author Response · Authors · 2024-12-02
> > **(12/2/24) Last day for us to respond**
> >
> > Hello reviewer Yi5e, we'd like to check in to see if there are any remaining questions or concerns before the author reply window closes.

---

### Official Review · Reviewer_QZxB · 2024-11-04

**Soundness:** 3
**Presentation:** 3
**Contribution:** 3
**Rating:** 6
**Confidence:** 3

**Summary:**

This paper trains a belief state transformer, that learns to predict the next token given the suffix and previous token given the prefix. The training is done jointly on both these objectives. The paper proposes that the training in this fashion can lead to better look-ahead when the goal state is present in the input representation. This is demonstrated with a simple planning task of star graphs where the model needs to choose a correct path to reach the goal state. Results show that this training results in a 100% success, whereas next token prediction results in chance performance. Further evaluation on tinyStories (with prefix) and unconditional generation is also performed showing superior performance.

**Strengths:**

The paper presents a new form of training that results in better belief state representation learning for LMs. This is shown with the performance on the star graph task where the belief state transformer achieves 100% performance. Similarly, the results on TinyStories also shows that this training regime achieves better LLM-as-Judge scores for completion of stories with prefix and suffix provided.

**Weaknesses:**

The task StarGraph seems to be very specific to the type of training objective proposed in the paper. The training involves learning the tokens left-to-right and right-to-left which seems aligned to the problem. Specifically, such an solution was mentioned in Bachmann & Nagarajan (2024), where the LM can learn to predict from "right-to-left" starting at the goal and ending in the start state. In order to claim that the proposed training objective results in a better belief state representation in general, would require more evaluations on planning problems like BlocksWorld etc. Specifically, showing the effectiveness on planning problems from benchmarks like PlanBench [1] and ACPBench [2] would make the claim of the paper stronger.

[1] Valmeekam, K., Marquez, M., Olmo, A., Sreedharan, S. and Kambhampati, S., 2024. Planbench: An extensible benchmark for evaluating large language models on planning and reasoning about change. Advances in Neural Information Processing Systems, 36.

[2] Kokel, H., Katz, M., Srinivas, K. and Sohrabi, S., 2024. ACPBench: Reasoning about Action, Change, and Planning. arXiv preprint arXiv:2410.05669.

**Questions:**

- For the StarGraph experiments, how was the backward latent $B(\emptyset)$ computed? How would the performance change if the training objective had $B$ but during inference $B(\emptyset)$ was not used?
- In the ablation section, what is the difference between `Belief w/o Backward` and single transformer next token prediction setup?
- Was this model trained on any other planning tasks that required look-ahead (for example stacking blocks to reach a goal state) and how did it perform in those cases?

---

> ### Author Response · Authors · 2024-11-16
> **(11/16/24) Response 1/2**
>
> Thanks for the review!
>
> > The task StarGraph seems to be very specific to the type of training objective proposed in the paper. The training involves learning the tokens left-to-right and right-to-left which seems aligned to the problem.
>
> We respectfully disagree with the point that BST seems designed to solve Stargraph here. BST is a general objective for sequence modeling motivated by belief state discovery, and this learning of belief states allows it to solve tasks that require lookahead, like the Stargraph.
>
> Conceptually, the BST objective is radically different when compared to the right-to-left solution. The right-to-left solution only uses the suffix to predict the previous token, which does not guarantee the discovery of belief states. For BST, both prefix and suffix embeddings are used to predict the previous token (as well as the next token), and this distinction is crucial for a belief state to arise (see Theorem 2).
>
> > Specifically, such an solution was mentioned in Bachmann & Nagarajan (2024), where the LM can learn to predict from "right-to-left" starting at the goal and ending in the start state
>
> The right-to-left solution is a task specific solution engineered for StarGraph, using domain knowledge for both training and inference. First, the solution exploits the structure of the stargraph by reversing the path data. Next, after training, the user must manually reverse the outputs to get the correct solution.
>
> In contrast, BST’s training is task-agnostic - we can train the BST without altering the data, and perform the standard left-to-right decoding to get the answer.  So while the BST and the right-to-left solution both solve the Stargraph and share a similar element of a previous token prediction term, the BST solution is more general and provably discovers belief states.
>
> > In order to claim that the proposed training objective results in a better belief state representation in general, would require more evaluation
>
> We provided formal proof that the Belief State Transformer training process converges to a compact belief state, and that methods like next-token and multi-token prediction do not. Could the reviewer point out a specific point of concern in the proof?  Additionally, we empirically verified on stargraph that the belief state transformer perfectly learns the correct belief state while the next-token transformer does not.

---

> ### Author Response · Authors · 2024-11-16
> **(11/16/24) Response 2/2**
>
> > Showing the effectiveness on planning problems from benchmarks like PlanBench [1] and ACPBench [2] would make the claim of the paper stronger.
>
> Showing BST’s effectiveness on more domains is valuable, as we have written in our limitations. However, showing a full-scale experiment before the end of the rebuttal period is difficult, and we kindly ask the reviewer to look at our current contributions:
>
> We proposed a new form of training for language models motivated by belief state discovery, showed that it overcomes hard planning problems known to confound standard approaches, provided theoretical proofs, and demonstrated good results in text generation.
>
> Our current contributions serve as proof-of-concept, and will hopefully inspire others to investigate the work further. The reviewer may be interested in our preliminary results in planning in a maze navigation setting, see below.
>
>
> > Was this model trained on any other planning tasks that required look-ahead (for example stacking blocks to reach a goal state) and how did it perform in those cases?
>
> We began experiments (after submission) of the BST on a 4-room maze navigation task, and found some promising results where BST greatly outperforms FIM in navigating to different goal locations. See below for some more details.
>
> We use the [four-room maze task](https://minigrid.farama.org/environments/minigrid/FourRoomsEnv/). In the four-room maze, the agent must navigate in a maze composed of four rooms interconnected by 4 gaps in the walls. We collect an offline dataset of 100K transitions using a uniform random data collecting policy, and train all models for 50 epochs.
>
> We evaluate the models on reaching goals of increasing difficulty. Type-0 goals are cells in the same room, Type-1 goals are cells in adjacent rooms, and Type-2 goals are cells in the most distant room. We compare BST with the FIM baseline on the mean success rate, results are listed as follows.
>
> |                 | Type-0 | Type-1 | Type-2 |
> | --------------- | ---------- | ---------- | ---------- |
> | BST        | **0.76**      | **0.53**       | **0.25**       |
> | FIM        | 0.23       | 0.12       | 0.00       |
>
>
> > For the StarGraph experiments, how was the backward latent B(∅) computed?
>
> We compute the backward latent by passing in the End-Of-Sentence (EOS) token, which is appended to every sequence in the stargraph training data.
>
> > How would the performance change if the training objective had B but during inference B(∅) was not used?
>
> The text head is conditioned on both forward and backward latents, so it is not possible to do inference without B(∅).
>
> > In the ablation section, what is the difference between Belief w/o Backward and single transformer next token prediction setup?
>
> The Belief w/o Backward still predicts next and previous tokens.

---

> ### Author Response · Authors · 2024-11-26
> **(11/26/24) Followup**
>
> Dear reviewer QZxB,
> We thank you again for your feedback. As the open discussion period draws to a close, we’d like to check back to see if you have any remaining concerns.
>
> We believe that we’ve sufficiently addressed your concerns and questions, but would be happy to clarify further. We would also be grateful for any other feedback.

---

> > ### Author Response · Authors · 2024-12-02
> > **(12/2/24) Last day for us to respond**
> >
> > Hello reviewer QZxB, we'd like to check in to see if there are any remaining questions or concerns before the author reply window closes.

---

### Author Response · Authors · 2024-11-26
**Global response (11/25/24)**

We thank the reviewers for their constructive feedback. We’ve incorporated suggested changes into the draft, with edits highlighted in blue. Below, we summarize all the draft changes for convenience; we have responded to each reviewer individually.

As requested by reviewer Aowt and Yi5e, we ran the Fill-in-the-Middle model on the stargraphs, and found that it performed poorly. The FIM model, like the forward transformer baseline, resorts to choosing a random path at test time. We believe this is due to the next-token prediction objective of FIM, which is known to fail on Stargraph. We updated figure 2 and section 3.4 with the results.

We then updated the text to be more explicit about experimental details. We updated the related work to mention other approaches like ELMo and BERT that incorporate future information. Finally, we added training overhead equations to the appendix.

---

### Meta-Review · Area_Chair_grf5 · 2024-12-23

**Metareview:**

The paper introduces the Belief State Transformer (BST), a novel architecture designed to improve goal-conditioned and non-myopic prediction by leveraging both prefix and suffix inputs to learn compact belief state representations. It demonstrates strong performance on tasks like pathfinding in star graphs and story generation, significantly outperforming standard next-token predictors and the Fill-in-the-Middle approach. The reviewers unanimously vote for acceptance with scores 6,6,6,8. The work is a strong proof-of-concept but could benefit from more extensive evaluations across diverse planning benchmarks. The main concern is that the evaluation is only on some toy benchmarks and lacks realistic evaluation on more challenging and modern tests. I think this paper is a nice addition to the community as a nice proof-of-concept work, and more evaluations on realistic tasks may be conducted in the future in larger-scale settings.

**Additional Comments On Reviewer Discussion:**

The reviewers appreciated the innovative Belief State Transformer (BST) for addressing limitations of next-token predictors and showing strong results on star graphs and story generation. Key concerns included the lack of large-scale evaluations, limited comparisons with bidirectional models like BERT, and the specificity of the tasks (e.g., star graphs) to the proposed objective. Reviewers also questioned why BST outperformed FIM, the scalability of BST, and clarity in proofs and experimental details. The authors addressed these by running additional experiments (e.g., showing FIM’s failure on star graphs), clarifying proofs and training overhead, updating related work to include BERT and ELMo, and highlighting BST’s task-agnostic nature. For the comparison with BERT-style models, the authors responded that BST still function in an autoregressive way at inference time, thus the main point is to compare against GPT-style approaches, which I think makes sense. The only concern I have is on the evaluation on broader and more realistic tasks, but I guess that may need more computing resources.

---

### Decision · Program_Chairs · 2025-01-22

Accept (Poster)